# Cognitive Support Technology for People with Intellectual Disabilities: Factors for Successful Implementation

**Michiel de Looze \*, Ellen Wilschut, Reinier Könemann, Kim Kranenborg and Harry De Boer**

TNO, 2316 ZL Leiden, The Netherlands; ellen.wischut@tno.nl (E.W.); reinier.konemann@tno.nl (R.K.);
kim.kranenborg@tno.nl (K.K.); harry.deboer@tno.nl (H.D.B.)
**\*** Correspondence: michiel.delooze@tno.nl

**Abstract:** In Europe, large numbers of people with disabilities are willing to work but have problems finding a job. One of the barriers to this is job complexity, particularly for those with low education, low IQ, or cognitive impairments. Digital technologies might help. Specifically, cognitive support technology (CST) has the potential to make jobs less complex and thus more accessible. CST may concern step-by-step digital instructions presented with monitors, tablets, smart phones, beamer projections, or near-eye displays. Based on cross-case evaluations, we aimed to define the success factors in the process of technology selection, development, and implementation. Four cases, situated at public social firms which offer jobs to people with disabilities, were selected. In each case, the optimal form of CST was selected. A qualitative analysis of subjective experiences of work accessibility, performance, usability, and acceptance was applied. The results were positive for most participants in most cases. Once installed, the CST was successful in simplifying jobs. A proportion of the workforce for which a specific job had been considered too complex was able to perform that job when supported by CST. Moreover, a majority of people judged the usability of the technology positively. For the consecutive steps of selection, development, and implementation, we ended up with eleven factors of success; these included, among others, shared and transparent decision making (in technology selection), the iterative and active involvement of workers to optimally adjust work instructions (in technology development), and explicit attention for psychosocial barriers (in technology implementation).

**Keywords:** digitalization; work instructions; AR support; cognitive disabilities; employability

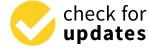



## 1. Introduction

The current labor market is tight in many countries. In most industry sectors, the rising number of vacancies is increasingly difficult to fill in. Meanwhile, many people with disabilities have no job and are struggling to find one. In the Netherlands, with a labor force of 9.9 million, the number of people who are willing to work but without a job exceeds 1 million (CBS 2023). The underlying reasons are diverse, but include, among others, low education, low IQ, low literacy, cognitive impairments, or psychosocial problems. The longer people remain unemployed, the more difficult it is to find a job. Those who manage to find a job often have difficulty keeping it (Blonk 2018).

In 2014, the European Union started its Cohesion Policy, of which the nineth priority was aimed at social inclusion (EU 2014). The Netherlands adopted the Participation Act in 2015. This law is aimed at creating jobs for people with a distance to the labor market, but it seems not to have had the desired effect (Van Waveren 2020).

A strategy to encourage more people to work is the use of digital technology. Specifically, cognitive support technology (CST) providing worker assistance on the job is likely to have potential in making jobs more accessible, especially where the complexity of tasks forms a barrier (Heinz et al. 2021). Impairments in memory, recall, information processing, decision making, or the ability to focus could be compensated for with CST (Wilschut et al.

2019). CST may concern step-by-step digital instructions that can be presented to workers through the means of monitors, tablets, smart phones, beamer projections, or near-eye displays (2D or 3D smart glasses). This presentation could be in the form of an overlay to the real world (augmented reality) or not (instructional reality). CST may also concern more general information provided through an app or another platform, to give support in the performance of tasks.

CST has been applied in the regular manufacturing industry to be more productive at high quality standards, and was reviewed by Egger and Masood (2020). Although positive experiences have been reported, other studies did not reveal significant differences in efficiency with using CST compared to paper-based instructions. Büttner et al. (2020) studied the effects of CST in learning an assembly task, showing that CST prevents assembly steps from being learned incorrectly.

Other studies have focused on the effects that CST could have for people with intellectual disabilities. Funk et al. (2015) showed that such workers assembled specific products up to three times faster and with up to 50% less errors when using projected instructions compared to electronic work instructions on a display. Korn et al. (2013) also reported higher productivity when using projected instructions. They also described a catalytic effect, showing a reduction in errors in most workers, but an increase in errors in some workers, which had been attributed to increased cognitive loading due to processing the digital instructions. Individual work capacity may have played a role here. In a follow-up study of Korn et al. (2014), the positive effects of projected instructions on productivity were not reproduced. Heinz et al. (2021) did report positive impacts, such as increased satisfaction and independence for the workers combined with higher productivity, following the implementation of CST in an assembly line. Bosch et al. (2020) described five cases of CST implementation studies performed in industrial practice, of which two concerned sheltered workplaces for workers with intellectual disabilities. These cases also showed mixed results. It is likely that factors other than the technology itself affect the outcome. These include the task characteristics and the worker's characteristics and capacities, but also the process of technology development and implementation.

In this paper, we describe four practical cases of selecting, developing, and implementing CST and evaluating its impact on people with intellectual disabilities. According to the DSM, this group of people is not only defined based on a low IQ, but also based on difficulties in conceptual, social, and practical areas of living (American Psychiatric Association 2022). In each case, it was the purpose to make jobs less complex, making them accessible for more people. Based on cross-case evaluations, we aimed to define the main factors of success in the process of technology selection, development, and implementation.

## 2. Methods

### 2.1. Cases

The cases were located at companies offering sheltered jobs to people with disabilities or placing these people externally. In the experience of these companies, the complexity of tasks is often a barrier to entering a job. Many of the workers within the sheltered companies perform simple work consisting of one or two actions, while work with multiple actions can only be performed by a small proportion of their work force. Therefore, these sheltered companies often cannot acquire the more complex work that regular companies are willing to outsource. In each of the use cases, we selected, developed, implemented, and evaluated a specific form of CST. Hereto, we aimed to follow the standard work plan as described below as closely as possible.

### 2.2. Standard Work Plan

Step 1 Preparation

We form a working group, consisting of company representatives like supervisors, job coaches, process engineers, (line-) managers, and experts in psychology, ergonomics, and

technology. In a kick-off meeting, we detail the work plan, discuss roles, responsibilities, and expectations, and make organizational arrangements.

Step 2    Analysis and selection

From work observations and interviews with workers and supervisors, we analyze the work process and tasks. Within tasks, we define the critical actions and the worker's needs for support. Next, we organize a workshop to discuss the needs of the workers and the company, the target group of users of the technology, and the options for technology solutions, as well as to search for the right match. The session is hosted by two experts in the field of CST and attended by two or three representatives of the company, preferably a manager as investor/decision-maker, an engineer as process owner and developer, and a worker as a potential end user. After the workshop, the company has more knowledge on the technologies and their potential to meet needs, have a prioritized list of suitable technologies, and have an idea of the involved costs and benefits. (Krause et al. 2022).

Step 3    Development and implementation

Following the selection of technology, a first application version is developed. Based on the support needs, the worker instructions and/or other assistive information are designed and programmed. These are tested and evaluated with workers and supervisors. From their feedback on the form and content, modifications are made. If necessary, more iterative cycles of testing and redesign follow with other workers in order to align with the needs of the target users. Technical implementation runs in parallel with these iterations.

Step 4    Evaluation

To evaluate the technology, we select the participants to be involved. Preferably, these participants do not have the cognitive abilities to perform the selected tasks without CST assistance, according to their supervisors. The participants are familiarized with the task and the technology.

The design of the test was dependent on the conditions of the companies involved, particularly in terms of the availability and willingness of people to participate, the availability of work (e.g., order intake), and the available time for testing.

Prior to the test, we assessed the potential of the technology to increase the accessibility of each selected task in each setting for each individual. Hereto, the competences of each individual were assessed by his or her supervisor. An ergonomics expert also assessed the task demands, both in cases of unsupported work and in cases of work supported by technology. The differences between the competences of the participants and the task demands in unsupported vs. supported work reflects the potential of the technology.

After the test, the experiences were collected from the participants and from their supervisors through questionnaires. The participants responded to the questions in an interview, while the supervisors filled in the questionnaire themselves. The items addressed in the questionnaires concerned the accessibility of the task, the usability of the technology, and the technology acceptance. Finally, we documented the process from steps 1 to 4. In specific sessions with company representatives, we evaluated the process, and discussed the lessons learned and the potential for further implementation.

## 3. Case Descriptions

*3.1. Assembly of an Electronic Unit (Case 1)*

### 3.1.1. Set-Up

Company A offers several types of work to people with intellectual disabilities (e.g., assembly work). The product to be assembled is a fuse box, an electronic unit consisting of a frame, connectors, and an electric circuit. The assembly task consists of 37 actions, e.g., picking, placing, tightening, and connecting wires, setting a tool, turning a mold, placing parts, and checking. Connecting the wires is a two-handed fine motor task. Employees check their own work, which is also a difficult step.

Beamer technology (Arkite) that can project step-by-step work instructions (text, symbols, and photos) onto the product, table, or trays of parts, as well as provide feedback when a wrong part is picked, was selected. The reasons for this were that both hands were needed during the work, feedback was required to ensure the work quality, locations of where to place the wires were critical for quality, and the system was used continuously during work shifts. These aspects make other types of hardware, like tablets or smart glasses, less useful. Also, projections on the product are more helpful then instructions on a display alone.

Twelve workers participated in the evaluation. They were trained in four two-hour sessions with the CST, with the supervisor's assistance gradually declining to zero. The training session was located in an area that was separate from the real production. After the final training session, those people that were considered to be able to work sufficiently independently and with sufficient speed and quality moved to the real production line with CST support (See Table 1).

**Table 1.** Overview of cases.

| | Case 1<br>Assembly I | Case 2<br>Assembly II | Case 3<br>Cleaning | Case 4<br>Order Picking |
|---|---|---|---|---|
| **general** | | | | |
| work | assembly of a product consisting of 37 actions, e.g., routing and connecting wires, using tools, and rotating the product | assembly of a product consisting of 24 actions, e.g., placing parts, routing of wires and quality control | multiple standard cleaning activities in various buildings of a school | picking orders consisting of 55 item types (clothes) from shelves into a picking bag |
| technology | stepwise in situ-projected work instructions | stepwise work instructions on a tablet | communication tool and platform on a tablet | stepwise work instructions on a smart glass |
| **process steps** | | | | |
| preparation | + | + | + | + |
| analysis and selection | + | + | + | +/−<br>work and needs analysis performed, no workshop to select the optimal system |
| development and implementation | +<br>multiple iterations of developing and testing with end users and team leaders | +<br>one iteration of developing and testing and feedback by team leaders | − | +/−<br>interface only marginally adjusted, no codesign with users |
| evaluation | +<br>12 subjects<br>2 h | +<br>13 subjects<br>2 h | − | +<br>22 subjects<br>2 h |
| **test set-up** | | | | |
| # subjects | 12 | 14 | − | 22 |
| background job of subjects | simple assembly work | simple assembly work | − | simple assembly (6), packaging (6) and diverse work (4) and order picking (6) |
| test duration | 2 weeks training and 6 weeks in real production | 2 h | − | 2 h |

+ indicates process steps performed as planned and described in Section 2.2. − indicates not performed. +/− indicates partially performed.

### 3.1.2. Outcome

The estimated potential to increase the accessibility of the task is illustrated in Figure 1. It shows the gap between the worker competences and the task demands. As expected, these gaps narrowed and approached zero in the tech-supported condition. On the other hand, the gap widened for reading (the instructions), while fine motor skills remained a problem that was not solved by the technology. In practice, it appeared that nine out of the twelve workers were able to learn to perform the task. The time required to assemble the product dropped from 30 to 15 min across the four two-hour sessions. When moved to the real production area, the product cycle times increased in the first week and then reduced to 10 min over time. After six weeks, however, we had seven dropouts. Thus, only two participants successfully completed the whole evaluation period. The transfer from a sheltered workplace to real production appeared to be problematic for most.

| | | without CST<br>competence score minus task demand | | with CST<br>competence score minus task demand | |
|---|---|---|---|---|---|
| cognition | comprehension | | -1.3 | | -0.3 |
| | memory and learning | | -1.6 | | 0.4 |
| | mental imagery | | -2.2 | | -1.2 |
| | problem solving | | -1.2 | | -0.7 |
| task performance | critical control | | -2.1 | | -1.1 |
| | organizing ability | | -1.0 | | -0.5 |
| | independence | | -1.2 | | -0.7 |
| | reading | | 1.2 | | 0.2 |
| physical | fine motor skills | | -1.9 | | -2.4 |

**Figure 1.** The difference in scores between the worker's competence score (averaged across subjects) and task demands for the situations with and without CST assistance in case 1. In case of a red bar, the competence is lower than the task demand.

The participants were rather positive about the usability of the technology. Quotes from the participants on the usability, both positive and negative, are included in Table 2. Most of them were positive in nature. When asked for individual preferences, nine participants indicated that they would prefer to learn and perform this type of work while supported by the technology; two preferred to work without the technology after the initial learning phase, while one person had no preference.

**Table 2.** Main outcome of the evaluations.

| | Case 1<br>Assembly I | Case 2<br>Assembly II | Case 4<br>Order Picking |
|---|---|---|---|
| **accessibility** | | | |
| technology effect on task demand vs. competence gap | Gap is reduced for all aspects of cognition and performance except reading gap unaffected for fine motor skills. | Gap is reduced for all aspects of cognition and performance except reading gap unaffected for fine motor skills. | Gap is closed, competences are estimated to outweigh task demands for nearly all aspects. Gap unaffected for fitness and stamina. |
| number of workers able to perform the task | Nine out of twelve were capable to learn the task and started to work in real production.<br>Two out of nine were successful in real production (7 dropouts). | Twelve out of thirteen were able to learn the task.<br>Five out of thirteen showed potential to work in real production, another 5 'possibly'. | Twenty-one out of twenty-four were able to learn the task. For two people, eye-related problems—and for one, limited intellectual ability—formed a barrier. |

**Table 2.** *Cont.*

| | Case 1 Assembly I | Case 2 Assembly II | Case 4 Order Picking |
|---|---|---|---|
| **usability** | | | |
| rating (1–10) | not measured | 7.8 | 7.9 |
| positive quotations | • 'easy, well-structured information' <br> • 'helps concentrating' <br> • 'helps remembering' <br> • 'makes me feel at ease' | • 'clear instructions' <br> • 'every time the same explanation' <br> • 'you can go back in the instructions' <br> • 'more extensive information is given' <br> • 'easy to follow the steps' <br> • 'coach is not always available, tablet is' <br> • 'don't have to remember all the information' | • 'easier to find the orders' <br> • 'you don't have to remember the location' <br> • 'quicker' <br> • 'handsfree' <br> • 'easy to learn and understand' |
| negative quotations | • 'rather have personal coaching' <br> • 'less control on how I want to do it' | • 'first time the tablet is useful for the second time I don't need it anymore' <br> • 'you have to search the information asking a colleague is quicker' <br> • 'I struggle with reading the text' | • 'difficult to work with because of my own glasses' <br> • 'you have to look up that's uncomfortable' <br> • 'it's easier without support' |
| **acceptance** | | | |
| preferences | Nine preferred the technology support. Two would have rather worked without this. One had no preference. | Nine preferred the technology support. One would have rather worked without this. Three had no preference. | Fourteen preferred the technology support. Four would have rather worked without this. Four had no preference. |

### 3.2. Assembly of a Fan (Case 2)

3.2.1. Set-Up

At Company B, various types of products are assembled by people with impairments. One of the products is a fan, which is meant to be mounted under a radiator to accelerate heat release. Due to its complexity, this work could be performed by a few workers only. Because of the increasing demand for this product, Company B wanted to involve more people in the assembly of the fan. Our evaluation focused on the first series of assembly steps that involved 24 actions, including picking, placing, conducting wires, connecting, and checking. The routing of the wires in the casing was particularly complex. Routing and connecting the wires required some fine motor skills. The workers used a small, thin stirring rod to the push wires in place.

The selected type of CST was a digital work instruction software platform (Azumuta) and tablet, on which the instructions were presented to the participants. The platform offers instructions to indicate 'how to perform an action', 'why this action is performed', and 'special points of attention', using text, photos, and videos. In this case, we selected this less expensive solution, because it was aimed at installing the technology on several workstations. At the same time, a broader implementation to other tasks was foreseen.

Fourteen workers participated in the evaluation. A short introduction by a supervisor to the assembly task was followed by an instructional video. Thereafter, the supervisor demonstrated in detail how to use the CST. The participants were asked to assemble the

products while assisted by the technology for two hours. They worked independently, but were able to ask for help from the supervisor in case they could not continue the work.

### 3.2.2. Outcome

Figure 2 illustrates the effect on the gap between the competences and task demands. Again, the gap was expected to reduce for a number of cognitive factors due to the increase in technology use; the technology created higher demands in terms of reading (i.e., reading the instructions), while fine motor tasks may remain a barrier for the participants unaffected by the CST. In the test, it appeared that two out of the fourteen participants were unable to perform the work, one due to a lack of capacity and the other due to a lack of motivation. It was estimated by the supervisor that four out of fourteen participants would be capable to work in the real production with technology support; five participants were not able to perform the tasks with enough speed or quality; and five participants showed potential, but more training time was needed to assess whether or not they could work in real production.

| | | without CST competence score minus task demand | | with CST competence score minus task demand | |
|---|---|---|---|---|---|
| cognition | comprehension | | -1.3 | | -0.3 |
| | memory and learning | | -1.6 | | 0.4 |
| | mental imagery | | -1.3 | | -0.3 |
| | problem solving | | -0.1 | | 0.4 |
| task performance | critical control | | -2.1 | | -1.1 |
| | organizing ability | | -1.0 | | -0.5 |
| | independence | | -0.6 | | -0.1 |
| | reading | | 1.0 | | -1.0 |
| physical | fine motor skills | | -2.3 | | -2.3 |

**Figure 2.** The difference in scores between the worker's competence score (averaged across subjects) and task demands for the situations with and without CST assistance in case 2. In case of a red bar, the competence is lower than the task demand.

The participants rated the usability of the technology with a 7.8 on average on a ten-point scale. Nine participants indicated that they would like to work with the technology in the future; one indicated that he would rather work without the technology, while three had no preference.

### 3.3. Cleaning (Case 3)

### 3.3.1. Set-Up

Company C is active in the cleaning sector, where they employ people with disabilities where possible. The company was interested in exploring the possibilities of technology to make cleaning work more accessible for people with disabilities. The location of this case was a school with multiple buildings. A team leader supervised ten workers with intellectual disabilities performing cleaning duties. She was at a distance from these workers for most of the time. The following bottlenecks emerged from the workshop: Firstly, the team leader found it difficult to judge the workers' emotional states at the start of the workday and act properly when the workers experienced frequent distress. Secondly, work-related communication during the day between the team leader and workers was difficult. The workers were spread around the building and were hard to find while mobile phones were often not answered or switched off. We proposed the following two technologies.

We proposed a communication tool (Whapbot, Game Solutions) in which an avatar (person on video) 'talks' with the employee. She welcomes him or her at the start of the day and asks how things are going. By clicking on symbols or emoticons, the employee can answer easily. Using a decision tree, the avatar can ask more detailed questions. The tool sends a signal to the team leader if action seems necessary.

Secondly, we proposed to mount a tablet with a collaboration software platform on the cleaning cart. The platform offers to assign and overview the progress of a task, share work instruction with photos and videos, and remotely communicate with the team leader.

### 3.3.2. Outcome

Unfortunately, the company decided not to continue the project. Therefore, the development, implementation, and evaluation steps have not taken place yet. The reasons for this decision were that the investments were considered too high compared to the benefits, and the company management desired a more high-tech solution like AR or VR.

### *3.4. Order Picking (Case 4)*

#### 3.4.1. Set-Up

Company D also offers several types of work to people with disabilities. The majority of workers carry out relatively simple packaging and assembly work. Some workers carry out order-picking work in a small warehouse.

For the order picking, paper order lists are used. The main challenges for the workers are not to become confused across the different order lines on the list and to pick the right number of products. On longer order lists, more mistakes are made that need reworking by the supervisor.

The technology tested was a smart glass (Augpick, Augmex). The smart glass shows step-by-step information on a small screen near the eye. In this case, the glasses show where the item is located in the warehouse, a photo of the item, and how many items you need to pick. Each picking step is confirmed by scanning the item barcode with a finger scanner.

#### 3.4.2. Outcome

The gap between the competences and task demands is expected to be eliminated by the technology for most aspects of cognition and performance. Fitness and stamina are the only aspects which might form barriers for the workers, as these competences are unaffected by the supporting technology (Figure 3).

| | | without CST | with CST |
|---|---|---|---|
| | | competence score minus task demand | competence score minus task demand |
| cognition | comprehension | -0.2 | 0.8 |
| | memory and learning | -0.2 | 0.8 |
| | mental imagery | -1.1 | 0.1 |
| | problem solving | -0.7 | 0.3 |
| task performance | critical control | -0.6 | 0.4 |
| | organizing ability | -0.6 | 0.4 |
| | independence | -0.1 | 0.9 |
| | reading | -0.2 | -0.2 |
| physical | fitness | -1.1 | -1.1 |

**Figure 3.** The difference in scores between the worker's competence score (averaged across subjects) and task demands for the situations with and without CST assistance in case 4. In case of a red bar, the competence is lower than the task demand.

In the evaluation, 24 workers participated. Six of them had warehouse experience, six worked in the assembly department, six in packaging, and four in various other departments. After a short training of no more than fifteen minutes, 21 employees were able to use the smart glass and perform the order-picking task independently. One participant from packaging was unable to work with the smart glass due to difficulty in understanding and following the instructions. For two participants, the technology was cased difficulty because of eye-related problems. For these workers, custom-made glasses could offer a solution in the future. It was estimated that 50% of the participants, who could not perform the task properly without technology support, were able to do so when supported. The us-

ability rating of the smart glasses (ten-point scale) was 7.9. A large majority, 14 participants, would prefer to work with the smart glasses in the future.

## 4. Discussion

### 4.1. Outcome of Cases

Our research indicates that the social services sector could benefit from the rapid developments in digitization to assist workers with intellectual disabilities. This paper focusses on one type of technology (CST) and its potential to make work more accessible for people with intellectual disabilities. The outcome of the evaluated cases was rather positive in most cases and for most participants. Our findings showed that the technology helped some of the participants to master specific jobs that were previously too complex to complete. Some of the participants for which the specific job had been considered to be too complex appeared to be able to perform the job when supported by technology. Moreover, a majority of people judged the usability of the technology positively. When asked for their preference, a large majority indicated that they preferred to work with the technology rather than without. The fact that they could continue to work with tech support independently from their supervisor was particularly liked.

Positive findings on cognitive support in the form of work instructions have been previously reported, in particular regarding their effect on the quality of work. Blattgerste et al. (2017), Funk et al. (2015), Uva et al. (2018), and Vanneste et al. (2020) reported significant reductions in errors made during assembly tasks when operators are supported by digital instructions. Some authors also reported positive effects of instructions on working speed (among others, Uva et al. 2018; Hou and Wang 2013), although opposite (negative) effects on working speed have also been reported (Blattgerste et al. 2017; Syberfeldt et al. 2015). Vanneste et al. (2020) concluded, based on a study in a population of people with disabilities, that digital work instructions had the potential to cognitively support operators and can hence contribute to better quality (less error), a lower stress level, a higher degree of independence, and a lower perceived complexity of the job. The same authors concluded that the non-significant effect on productivity seems to be moderated by the cognitive skills and experiences of the population.

In our cases, the outcome was also not uniform. Not all participants were enthusiastic. A few people had a preference for the support of a human supervisor over the technology and other participants dropped out untimely, mostly due to issues related to their vulnerability. In one company, the technology proposed was not approved by higher management, and thus, the project stopped before the technology was developed and installed.

Based on these outcomes, we defined the success-contributing factors.

### 4.2. Limitations

Ideally, to attribute or correlate the outcome of our evaluations to process factors, the cases should involve some variation in the implementation process, with other factors being relatively constant. In our study, the processes of development and implementation showed some variation despite some overlap. Other factors, however, were not constant, as the cases concerned different companies, tasks, and workers, while the test protocols differed in terms of the test duration and number of participants.

Another limitation relates to our selection of cases, for which we were quite dependent on the willingness of companies to collaborate. It is possible that the inclusion of other cases would have led to other or additional success factors. Another limitation of the present approach is that the long-term effect of the implementations was outside our scope. Therefore, it cannot be determined whether the performance and short-term experiences of people on the usability and acceptance would deteriorate or improve in the long run.

Similarly, the number of participants was limited. The total number of participants in testing the technologies was 44.

Despite the above issues, we believe that our findings and experiences in the four cases of product development resulted in sufficient clues to compose a list of factors at the heart

of successful technology development and implementation. We have grouped these factors into the sequential process stages of technology selection, development, implementation, and testing and further implementation, as shown in Figure 4.

| selection | development | implementation |
|---|---|---|
| - clear understanding of working conditions, work process and tasks and human needs for support | - acknowledgement of importance of high-quality working instructions in terms of content and form | - good introduction of the technology and training of workers |
| - transparant decision-making based on shared understanding of aim, target users, worker's needs, and pros and cons of technologies | - active involvement of workers to ensure optimal adjustment of worker instructions to users | - explicit attention for psychosocial barriers for successful employment |
| - commitment of decision-makers from the early beginning | - application of iterative cycles of developing and testing the work instructions | - understanding of a modified role of supervisors due to the technology |
| | | - organizing the capacity to master the new technologies |
| | | - wide commitment within the organization and willingness to invest |

**Figure 4.** Key factors of success in selecting, developing, and implementing cognitive support technology.

*4.3. Key Factors of Success*

4.3.1. Selection

In the selection of the type of technology, we opt for a human-centric approach. Rather than technological feasibility or efficiency considerations, the people and their cognitive, social–emotional, or physical needs for support should form the starting point. In ergonomics, the human factor has always been the key point, but currently, human centricity is receiving renewed attention within other disciplines and innovation agendas like Industry 5.0. Within the human-centric approach, we distinguish the following success factors:

1.  One needs to have a clear view of the working conditions, the work processes and tasks, and the human needs for support. The task complexity as perceived by workers is an important aspect herein, also discussed by Wiedenmaier et al. (2009). Different conditions, processes, tasks, and needs may require different types of technology. For instance, smart glasses could be advocated for where workers are mobile, need two hands for task performance, and do not need too much information. Where workers are stationary and a lot of information is needed, on-site projection technology may be attractive. Touch-screen displays and tablets on which digital work instructions are communicated may work well in cases where multiple workstations need to be equipped, provided that these can be placed within view and their operation does not interfere with the performance of a two-handed job.

2.  The reasons for selecting specific technologies should be clear to all involved. The selection decision should be transparent and based on common understandings. This holds for the aim. CST may serve different aims like increasing productivity (work speed), increasing quality (reducing error), or increasing accessibility for 'weaker' groups. The aim could be to learn new jobs faster or to provide continuous support in daily operations. Consensus is not always self-evident. We also need common understanding on the target group of users of the technology, the worker's needs, and in reaction to these needs, the pros and cons of different technologies. The organization of a dedicated workshop, such as that involved in our work plan (see Section 2.2), could be helpful to realize the above.

3.  Decision-makers should be included in an early stage. One may involve them in the dedicated workshop, where they could be informed and could contribute to the discussions. In our cases, decision-makers attended to the workshop or were properly informed from the beginning, except for in the cleaning case. In this case, we learned that a more expensive high-tech solution (like VR or AR) is not always necessary, and

that a relatively simple solution that requires little investment can already bear fruit. Remarkably, this outcome did not meet the expectations of the decision-makers and thus the project stopped untimely.

### 4.3.2. Development

After selection, the stage of development starts. For CST, this refers to the making and the programming of step-by-step working instructions in one of the already existing platforms. The success-contributing factors in this stage are the following.

1.  The importance of a good working instruction cannot be underestimated, which has been argued before by Söderberg et al. (2014). One should be selective in the amount of information, not giving too little nor too much (Krause et al. 2022). Of course, the instructions should be easily understandable for the target users. It could be helpful to develop several levels of information, so one can adapt to the varying capacities of individuals. It certainly helps when the screen layout is consistent in where to show what. The amount of text should be kept to a minimum. Good and sufficiently detailed pictures (with annotations) or even short videos could be preferable above text blocks.

2.  The active involvement of workers to ensure that worker instructions really meet the worker's requirements is crucial. As much direct worker participation as possible has been long advocated for within occupational interventions and is one of the main issues in Participatory Ergonomics (Noro and Imada 1992). This is based on the notion that technology developers cannot oversee the consequences and impact of their design decisions on the end users of the technology. This notion might even be stronger for workers that are intellectually disabled in one way or another, although obtaining meaningful feedback from these workers is more complex.

3.  The application of iterative cycles of development and testing with the end users is highly recommended. Iterative testing was applied in the assembly cases only, not in the order-picking case. Most meaningful feedback was retrieved after observing the end users while using the technology and interviewing them on the observed issue and deviations from instructions. Iterative testing in the assembly cases led us towards detailed refinements, as well as the decisions to make separate instructions for left-handed and right-handed people and two separate instruction levels: one for beginners and one for those who only needed part of the instructions after some experience. In the order-picking case, we just implemented the basic version provided by the technology producer. Hence, in this case, we ended up with some design recommendations, only after our evaluation.

### 4.3.3. Implementation

Once developed, the technology can be implemented within the company. The factors to address at this stage are the following:

1.  The technology should be properly introduced to the workers. Vanneste et al. (2020) argue that when the supportive technology is new to workers, it can affect their performance, which can be alleviated by taking the time to explain its functioning at the beginning. Our experience with public social firms has found that a short explanatory introduction to workers with intellectual disabilities can be sufficient. This often works best with a follow-up, whereby the worker takes their time with the technical support to master the task. Supervisors do not need to stay close by, but should be available in case of questions. The dependency of workers on supervisors depends on the design of the interface. For instance, an initial video presenting the work process on a more general level before diving into each consecutive step might reduce the supervisor's task.

2.  Cognitive support technology handles the technical explanation and guidance of the execution of the required working activities. With this target group, various factors other than the complexity of the task execution, mainly of a psychosocial nature, may

stand in the way of performing well in the new task. If not addressed, the psychosocial factors may jeopardize all efforts to teach the technical task executions

3. The role of the supervisor will change due to the technology. The technical explanation of the task execution and the training of it can be carried out through the technology for a significant part. It is clear, however, that the supervisor still needs to be there to support workers psychosocially. It is recommended to consider the modified role of the supervisor in the implementation stage.

4. The adopting company should have or organize the capacity to master the new technologies. This includes the making of work instructions, but also the technical installation of hardware and the integration of CST software platforms into information or data-management systems that are used in the company.

5. Finally, a wide commitment within the organization and a willingness to invest in terms of human capacity, time, and money in CST implementation are required to fully benefit from its potential.

## 5. Final Considerations

CST has the potential to increase the work participation of people with disabilities within the labor market. It may reduce task complexity as a barrier to entering a new, more complex job. Moreover, it may empower people and encourage personal autonomy, as people can perform more work with added value. Meanwhile, they are less dependent on a job coach or supervisor. The benefit for the supervisor is that his difficult and time-consuming task of verbally providing the required and repeated instructions to the workers is mediated now with the technology. As a consequence of these benefits, more complex work could be outsourced from regular companies to public social firms. Meanwhile, if applied in regular companies, CST may support workers to move from sheltered work at a public social firm to a 'real' job in a regular company.

However, the use of CST is not widespread. The wide communication of the pilot results in terms of effectivity and implementation factors contributing to success might help. Creating impact in terms of increased work participation requires the common effort of multiple stakeholders, including public social firms, employers in regular companies, technology developers, and implementation experts. Moreover, public authorities and agencies involved in finding the right job for each job seeker should be aware of the potential of the technology. A specific job that is considered to be too complex for a specific person can become feasible with technological support.

In conclusion, our pilots, in which we focused on different types of work and different types of CST, showed the wide potential of CST for people with cognitive abilities. They also indicated the importance of a good introduction and implementation in the companies. This paper ended up with the formulation of several factors of success for the various stages of implementation, which could be of help in accelerating future CST adoptions.

**Funding:** This research was funded by Goldschmeding Stichting voor Mens, Werk en Economie, grant number 33 and by Topsector Life Sciences and Health, grant number 201910130.

**Institutional Review Board Statement:** The study was conducted in accordance with the Declaration of Helsinki, and approved by the Institutional Review Board of TNO (protocol code 2022-85 and 20/1/22.

**Informed Consent Statement:** Informed consent was obtained from all subjects involved in the study.

**Data Availability Statement:** Not applicable.

**Conflicts of Interest:** The authors declare no conflict of interest.

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
