# Peer review of "Cognitive Support Technology for People with Intellectual Disabilities: Factors for Successful Implementation"

_socsci, doi:10.3390/socsci12110622_

Round 1

Reviewer 1 Report

Comments and Suggestions for Authors

In general, the article is very coherent, well structured and presents the subject matter in a clear and precise manner, the case study is presented clearly and is understood from beginning to end. The part that is missing is a more extensive discussion, contrasting with the opinion of other authors, as well as with similar studies.

Finally, a concluding section would also be appropriate in which the main findings are highlighted.

We suggest that the authors extend the discussion section and contrast it with other authors and similar cases and that they add a section of conclusions with the main ideas obtained.

Congratulations for your contribution.

Author Response

  1. We incorporated the main findings from similar research in the discussion (highlighted in yellow in the discussion section)
  2. We added a short paragraph on the conclusions at the end of the paper. Actually we considered the formulation of the success factors as a sort of conclusion. Upon your request we now have added a short paragraph in which we only put these factors (without repeating them) into the right context and therefore chose to keep this paragraph relatively short (if you agree).

Reviewer 2 Report

Comments and Suggestions for Authors

 This is a clearly written manuscript, which I found easy to read. This is a rarity in academic writing. I am familiar with the learning aids for people with intellectual disabilities described in this study and the research accurately captures the benefits. It was an interesting study that deserves publication. The conclusion was excellent. There are, however, some minor yet important issues that need addressing before meeting publishing standard.   

Don’t use ‘impaired’. ‘Impaired people’ is even worse. Ask yourself, what is an impaired person? Someone with an injured foot is impaired. 

Always put the person first – ‘people with disabilities’

Don’t use the term ‘vulnerable people’. It’s ambiguous and a little condescending. For example, refugees are vulnerable people. Be specific to avoid confusion.  

‘cognitively impaired’ – should be ‘people with intellectual disabilities’. 

You need to define intellectual disabilities. Cite the DSM-5. It is not just IQ but also social and self-care functioning.

You also need to explain the benefits of the technology. For example, people with intellectual disabilities often require repeated instructions to perform tasks – which is difficult and time-consuming for a supervisor. The technology mediates this difficulty because the worker can replay the instructions.

‘poor education’ – should be ‘low education’

Referencing needs consistency. APA formatting has a comma after the author. This is correct (European Commission, 2021)

Participation act - Participation Act

Past tense is needed in some instances – e.g., ‘detail’ should be ‘detailed’

‘Company A offers several types of work to impaired people, e.g. assembly work’ – Should be ‘Company A offered several types of work to people with intellectual disabilities (e.g. assembly work)’.

Numerals – all numbers up to ten are spelt out according to APA formatting rules. They become numerals after 11 unless the number begins a sentence (e.g., ‘12 workers participated …’ – ‘Twelve workers participated’).

‘technology, was expected’ – no comma.

‘The cleaning work was carried out by 10 vulnerable workers supervised by a team leader operating at distance from workers for most of the time’ – vulnerable? Suggested edit – ‘A team leader supervised ten workers with an intellectual disability performing cleaning duties most of the time’.

Lines 231 to 233 should be: ‘Firstly, the team leader found it difficult to judge the workers’ emotional state at the start of the workday and to act when the workers experienced frequent distress’. On another note, this indicates incompetence on the team leader’s part. Their poor management, and possibly bullying and/or exploitation, is likely causing the distress. It might explain why the phones were turned off.

(from packaging)  - don’t use brackets here.

300-301 should be: ‘Our research indicates that the social services sector could benefit from the rapid developments in digitization to assist workers with intellectual disabilities’

305 to 307 should be ‘Our findings showed that the technology helped some of the participants to master specific jobs that were previously too complex to complete’.  

312 – enthousiastic – enthusiastic

325 to 333 should be one paragraph.

Sentence in line 333 doesn’t make sense (i.e., 12 to 18)

Comments on the Quality of English Language

Some clumsy sentences need clarity. These are mostly to do with the incorrect use of a comma. Here is an example:

"In our experience in the public social firms, a short explanatory introduction can be sufficient, followed by a period in which people take their time to learn the task with technical support".

Should be 'Our experience with public social firms has found that a short explanatory introduction to the worker with an intellectual disability can be sufficient. This often works best with a follow up whereby the worker takes their time with the technical support to master the task.

I also noted some incorrect use of tense, which should be past tense. It's just a matter of doing another proof read. 

Author Response

Thanks for your helpful suggestions. We followed all your suggestions resulting in the high-lighted text changes.

Regarding your remark on past vs present tense, I would like to indicate that in 2.2 we use present tense, because it is the description of our standard work plan, and not a description of actions we have performed in the specific pilots. (Actually we sticked as much as possible to this plan but we ended up with some deviations and variations across cases as described). That is the reason why we use present tense there, in contrast with the rest of the paper (which I have checked for mistakes).

Regarding your remark: Lines 231 to 233 should be: ‘Firstly, the team leader found it difficult to judge the workers’ emotional state at the start of the workday and to act when the workers experienced frequent distress’. On another note, this indicates incompetence on the team leader’s part. Their poor management, and possibly bullying and/or exploitation, is likely causing the distress. It might explain why the phones were turned off.

According to the  first part of your suggestion we modified the phrasing, The second part (from on another note) is quite speculative.  In my view it was rather the hectic situation at start, very busy in the school and too much happening when workers arrive and also splitting up really fast after arrival. But this is also quite speculative, therefore I did not add this to the text.